# Learning Italian as a Second Language in a Sample of Ukrainian Children: A Game-Based Learning Approach

Alessandro Frolli * , Francesco Cerciello , Clara Esposito, Mariagrazia Russo and Fabio Bisogni

Disability Research, Centre of Rome University of International Studies, 00147 Rome, Italy; francesco.cerciello@unint.eu (F.C.); clara.esposito@unint.eu (C.E.); mariagrazia.russo@unint.eu (M.R.); fabio.bisogni@unint.eu (F.B.)
* Correspondence: alessandro.frolli@unint.eu; Tel.: +39-347-4910178

**Abstract:** Game-based learning is an educational approach aimed at acquiring knowledge through the use of play. There are various studies that note the effectiveness of playing as an educational tool and the use of digital platforms as a tool that can increase its effectiveness. We wanted to investigate whether a game-based learning approach may determine an improvement in Italian as a foreign language in terms of vocabulary expansion. The sample consists of 48 Ukrainian children between 6 and 7 years old. There were 24 female children and 24 male children in the sample, divided into two groups: the control group (Gr1) learnt Italian through frontal lessons (traditional approach), the experimental group (Gr2) learnt Italian through games and activities (game-based learning). The results have shown that the experimental group had a major increase in Italian vocabulary. However, both groups had an increase in this sense. Game-based learning remains an effective and promising educational approach, but other variables must be taken into account. Furthermore, the scarcity of literature on foreign language learning through game-based learning creates the need for more studies.

**Keywords:** game-based learning; language learning; vocabulary; learning; digital game-based learning

## 1. Introduction

Games are widely recognized as a conventional activity and a captivating experience for children, adolescents, and adults. The elements of games can encompass various aspects, including mechanical actions that entail repeated participation in essential tasks by the player, visual appeal, storytelling elements, power-ups, rewards, and auditory enhancements [1]. In general, games or their constituent parts are viewed as beneficial [2,3]. As an illustration, King et al. [4] observed that incentives, such as earning points, constituted the most enjoyable and significant aspects of engaging in video game play. The pedagogical approach of employing games as a tool for learning is known as game-based learning. The integration of gaming into educational practices has resulted in the development of numerous inventive ideas. Through the introduction and utilization of serious games, educators now have alternative approaches to enrich the process of teaching [5]. In contrast to more traditional educational approaches, game-based learning primarily involves incorporating different game components into non-game contexts. This strategy aims to captivate participants and boost their enthusiasm. The scope of this educational approach is broad, encompassing both non-technological and technological elements of games that are integrated into teaching activities [6].

Game-based learning environments are believed to foster interest, motivate learners, facilitate instruction adherence, and ultimately enhance learning outcomes [7,8]. Spires et al. [9] describe game-based learning as more than simply creating games for students; instead, it involves designing interactive educational activities that lead students toward intended goals. Furthermore, Kapp [10] (p. 23) defines a game as "a system that engages

players in an abstract challenge defined by rules, interaction, and feedback, resulting in a quantifiable emotional outcome". Gamification is described as the "deliberate and strategic application of game theory to problem-solving that promotes learning and incorporates an emotional component" [10] (p. 12). Stiller and colleagues [11] demonstrated that employing a playful teaching method increased students' motivation, participation, and cognitive abilities. In recent years, mounting evidence has shown that game-based learning can enhance student engagement, boost their motivation to learn, and increase the effectiveness of teaching [12–15]. It is a commonly held belief that in order to obtain a more comprehensive comprehension of the inner workings of game-based learning tasks and to draw valid conclusions, additional research is required, specifically in the form of extensive randomized control trials [2,7]. In this approach, the games employed can be of various types, and the selection depends on the desired objective and the subject matter to be taught.

Today, technology is an omnipresent tool, and the utilization of digital games as a teaching tool cannot be overlooked. When designed with the learner at the forefront, digital game-based learning can indeed be an effective educational tool aimed at improving learner interaction, cooperation, and communication. Moreover, as demonstrated in Casañ-Pitarch R.'s [16] study, the use of mobile phones, tablets, and computers allows students to engage in both individualized and collaborative learning experiences within the classroom or at home. Anastasiadis and colleagues [17] found that digital game-based learning aided in the development of higher self-esteem, accelerated the acquisition of soft skills, improved critical thinking, and honed problem-solving abilities. One particularly useful tool for game-based practices in recent years has been the digital platform Kahoot! This platform was the focus of a recent study conducted by Ahmed and collaborators [18], which highlighted the usefulness of employing such games for vocabulary acquisition and practice in second language (L2) teaching. The study demonstrated how the use of this platform trained and enhanced learners' capacity to acquire and retain new lexical items. Additional research [19] makes reference to foreign language acquisition in terms of lexicon expansion through the utilization of gaming activities, such as software and computer-based programs.

Continuing in a similar vein, Tsai et al. [20] assert that the incorporation of games into foreign language learning holds significant implications for students' achievement, with the potential to enhance collaboration and motivation. As such, games exert a favourable influence on the language acquisition process by mitigating student apprehension, providing entertainment, and affording learners the prospect of engaging with novel methodologies for acquiring the target language, deviating from conventional pedagogy [21]. Currently, there is a growing trend in utilizing game-based learning, which is being widely implemented in various fields of study. One such field is medical education, where it has been observed that students in this discipline are often receptive to new and innovative approaches, especially those involving modern technologies. As a result, there has been an increased use of interactive and captivating games in medical education [22,23]. Although the number of studies investigating game-based learning is increasing, research on the effects of this approach specifically in language learning remains limited. Let us posit that a game-based learning could potentially facilitate the processes of acquiring Italian as a foreign language. It is anticipated that the expansion of vocabulary would be more pronounced within the cohort engaged in gaming activities, in contrast to the group undergoing conventional pedagogical methods. Consequently, our objective is to define and examine the impact of experiential play (both traditional and digital games) in the teaching of Italian as a second language (L2) to a group of Ukrainian children.

## 2. Materials and Methods

The "Parliamone" project, which took place at the Rome University of International Studies between June and July 2022, served as the source for the sample under investigation. The primary objective of this project was to provide Italian language instruction to Ukrainian refugees over the course of 8 weeks. The sample comprised 48 Ukrainian children between 6 and 7 years old, who had arrived in Italy one month prior to data

collection. In order to ensure homogeneity, the inclusion criteria were as follows: (a) IQ within the normal range, assessed using the Coloured Progressive Matrices of Raven [24]; (b) absence of psychopathological disorders, as determined by the Kiddie Schedule for Affective Disorders and Schizophrenia (K-SADS) [25]; and (c) absence of socio-cultural disadvantages. To ascertain their cultural class, the Economic Social Scale (SES) [26] was employed. The sample consisted of 24 female and 24 male children, with an average age of 6.5 (SD 0.50) and an average SES of 7.2 (SD 0.59). The sample was divided into two groups (see Table 1): the control group (Gr1), consisting of 24 children who received Italian instruction through frontal lessons utilizing a traditional approach, and the experimental group (Gr2), consisting of 24 children who learned Italian through games and activities employing a game-based learning approach. The data were collected in collaboration with the Italian Foundation for Neuroscience and Developmental Disorders (FINDS), and qualified psychologists gathered the information at the Disability Research Centre (DRC) located at the Rome University of International Studies (UNINT). The data collection spanned a period of four weeks, during which the children were exposed to three hours of Italian lectures per day, twice a week (six hours per week), facilitated by MSc students from the Rome University of International Studies.

**Table 1.** Description of the sample.

|  | **All** | **Gr1** | **Gr2** |
|---|---|---|---|
| N | 48 | 24 | 24 |
| Male (number) | 24 | 12 | 12 |
| Age (years) | 6.50 (0.50) | 6.35 (0.49) | 6.45 (0.45) |
| Sociocultural background (SES) | 7.20 (0.59) | 7.35 (0.40) | 7.15 (0.65) |
| Comorbidities (K-SADS) | Absence | Absence | Absence |
| Intelligence (CPM) | 33.46 (1.08) | 32.38 (1.09) | 34.43 (1.08) |

As shown in Table 1, the two groups are comparable in terms of gender, age, socioeconomic background, and absence of psychopathologies.

*2.1. Instruments*

The protocol employed for evaluating the inclusion criteria and managing the study encompasses the following assessments:

1. Socioeconomic Status (SES ): A questionnaire to gather information on the educational and professional backgrounds of parents, as well as their social positioning. This self-administered socio-demographic questionnaire aimed to collect data on the Socioeconomic Status (SES)of each parental pair, using the Hollingshead index for calculation. Along with other demographic and lifestyle variables, the questionnaire aimed to create a composite index that factors in both the parents' level of education and their occupational status. The range of scores for the SES index is as follows: (a) 0 to 3 represents a low social class, (b) 4 to 5 indicates a medium level, and (c) 6 to 8 signifies a middle-high level [26];

2. MacArthur–Bates Communicative Development Inventory (CDI): The Communication and Language Development Inventory (CDI) is a questionnaire that aims to evaluate the communication skills of young children as they progress from basic comprehension and non-verbal communication to the expansion of vocabulary and the early stages of grammar. This assessment tool provides benchmarks based on a sample of typically developing children. The CDI is available in two versions, presented in a checklist format: (a) the Words and Gestures version, designed for children aged 8 to 16 months, and (b) the Words and Phrases version, intended for children aged 16 to 30 months. The Words and Gestures version includes assessments of pre-linguistic abilities, such as responding to their name, using verbal tags, and

imitating sounds. It consists of 28 sentences, and caregivers are asked to indicate whether their child understands each sentence. Additionally, it includes a list of 196 vocabulary items, allowing caregivers to specify whether the child "understands" or "understands and says" each word. Additionally, it includes 63 gestures categorized into five groups, including early gestures associated with social engagement and later gestures involving actions, play, and object-directed imitation. The Words and Phrases version comprises 680 vocabulary entries, and caregivers only indicate whether the child produces each item, without referencing understanding. The second part of the questionnaire covers various grammatical elements [27];

3. Kiddie Schedule for Affective Disorders and Schizophrenia (K-SADS): The Kiddie-Schedule for Affective Disorders and Schizophrenia (K-SADS) is a diagnostic interview utilized to evaluate past and present psychopathological disorders in children and adolescents aged 6 to 17 years, based on DSM-5 criteria. It encompasses multiple components, including: an introductory unstructured interview, a diagnostic screening interview, a checklist for administering diagnostic supplements, and five diagnostic supplements for mood disorders, psychotic disorders, anxiety disorders, attention deficit disorders, and disruptive behaviour, as well as substance abuse. Each of these supplements provides the necessary DSM criteria for the respective disorders being assessed. Additionally, the K-SADS incorporates a comprehensive checklist of the patient's clinical history and a scale for an overall assessment of the child's current functioning, known as Von Zerssen's global functioning scale (VGF) [25];

4. Coloured Progressive Matrices of Raven (CPM): A test used to assess the intelligence of children aged 4 to 9/10 years. It involves a series of cards, where participants are asked to complete missing figures within each card. The test includes the A and B matrices from the standard test, along with an additional AB test comprising 12 items. Most items in the matrices are coloured for enhanced visibility, except for the final items in the B series, which are written in black ink on a white background [24].

### 2.2. Procedures

After the evaluation of the inclusion criteria and the division of the sample into two groups, a pretest (T0) was conducted using The MacArthur–Bates Communicative Development Inventory (CDI) to assess the children's baseline level of Italian language proficiency in terms of vocabulary size (number of words). Subsequently, the children underwent Italian language instruction for 3 h per day (twice a week), 6 h per week, delivered by trainees and language students from the Rome University of International Studies. The control group (Gr1) received teaching activities based on a traditional approach. These activities focused on various aspects of the Italian language, including the alphabet, numbers, colours, frequently used words, basic grammar, and more. The instruction was primarily delivered through oral lessons with the support of a blackboard and PowerPoint presentations. On the other hand, the experimental group (Gr2) also covered the same teaching units as the control group, but employed via a game-based approach. The topics covered in the lessons progressed in difficulty, starting from basic vocabulary such as the alphabet, greetings, food, colours, and numbers, and gradually advancing to more complex grammar topics such as nouns, determinative and indeterminative articles and, finally, the three conjugations of regular verbs in the present tense. Each explanation was accompanied by a visually engaging PowerPoint presentation with vibrant colours and images to aid the children's comprehension and enhance their engagement. To reinforce the concepts taught, a game phase followed each lesson, eliminating the need for traditional lecturing. The game phase took place outdoors on Friday afternoons, and games were chosen or specifically created to align with the topics covered in the lessons. For example, one of the games played was 'dunk-seven', where children dribbled a ball while counting to 7 and attempted to dunk the ball on the seventh dribble to eliminate someone from the game. The last person remaining in the game was declared the winner. For the game phase related to grammar topics, interactive quizzes called "kahoots" were created. These quizzes not only

provided entertainment and engagement for the children, but also served as a valuable assessment tool for the teachers. The teachers could evaluate the number of correct and incorrect answers for each question without identifying the individual respondents. At the end of each "Kahoot!" session, the students who accumulated the highest number of correct answers and achieved top scores received red stars as rewards. In our opinion, based on the observation of children behaviour, this incentive system fuelled the students' motivation to win and encouraged them to pay closer attention during the explanations, leading to better comprehension and improved performance in subsequent kahoots. During the explanation phase, the primary language of instruction was Italian, although Russian-speaking teachers were present. The Russian language was only used in cases of extreme necessity or during PowerPoint presentations of more complex grammatical topics that did not exist in Russian or Ukrainian grammar. For instance, the explanation of articles, which are absent in Russian and Ukrainian grammar, required the occasional use of Russian to aid comprehension. The final day of the lessons was dedicated entirely to games. A specially designed 'Snakes and Ladders' game titled 'The Italian Olympics' was created, incorporating questions related to the topics covered throughout the course. The questions progressively increased in difficulty. The first three children to reach the finish line received giant soft toys as prizes, while all other participants were able to select a prize of their choice later. At the conclusion of the project, the MacArthur–Bates CDI was reassessed (T1) to determine the increase in Italian vocabulary size (number of words) among the children.

### 3. Results

The data analyses were conducted using SPSS 26.0 statistical survey software. Statistical significance was set at the 5% level ($\alpha < 0.05$). In this study, we utilized an ANOVA to compare the scores obtained from measuring the number of words learned from T0 to T1 and between the two groups. We referred to the first group as Gr1, which underwent traditional learning training, and the second group as Gr2, which underwent game-based learning training. T0 represented the measurement taken before initiating the learning training, while T1 represented the measurement taken after the training, which lasted for 4 weeks. To be more specific, we compared the performance of both groups (1 and 2) at T0 and T1 by analysing the number of known words (vocabulary) to determine which intervention was more effective (time–within variable). Additionally, we compared both groups at T1 to examine the differences between them (group–between variable). Consequently, we performed a 2 × 2 mixed repeated measures ANOVA, with the within factor being time (T0 and T1) and the between factor being group (Gr1 and Gr2).

The analyses showed the following results:

- Time × group interaction is significant [F (1.46) = 219.215, $p < 0.05$]. This finding indicates that there is a significant interaction between time and intervention type. More specifically, both trainings showed an effectiveness expressed through a higher number of learned words, but this effectiveness was more significant in Gr2 (Table 2 and Figure 1).

**Table 2.** Time × group interaction.

| Time | Group 1 | | Group 2 | | F | *p* |
|---|---|---|---|---|---|---|
| | **Mean** | **SD** | **Mean** | **SD** | | |
| T0 | 40.54 | 4.62 | 40.29 | 4.19 | | |
| T1 | 139.29 | 4.80 | 171.91 | 5.45 | 219.215 | <0.05 * |

* Statistical significance.

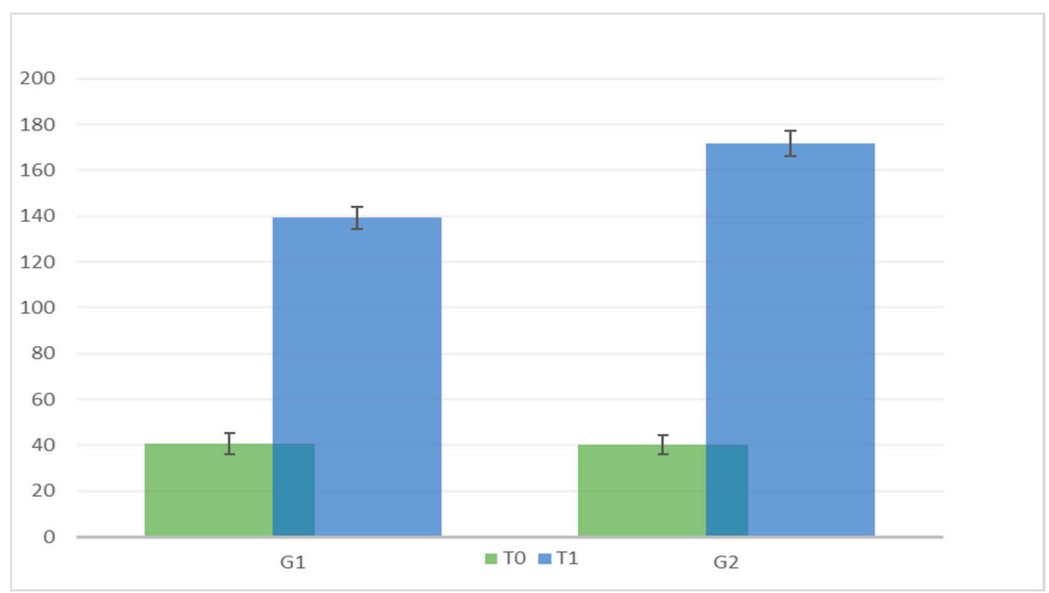

**Figure 1.** Comparison of G1 and G2 at T0 and T1.

In Tables 3 and 4, the effect sizes are shown. In particular, Table 3 refers to effects size within subjects, whereas Table 4 effects size between subjects.

**Table 3.** Effect size within subjects.

| | | Sum of Type III Squares | df | Quadratic Mean | F | *p* | Partial Eta Square |
|---|---|---|---|---|---|---|---|
| Time | Sphericity | 318,435,844 | 1 | 318,435,844 | 18,493,552 | <0.001 | 0.998 |
| | Greenhouse-Geisser | 318,435,844 | 1000 | 318,435,844 | 18,493,552 | <0.001 | 0.998 |
| | Huynh-Feldt | 318,435,844 | 1000 | 318,435,844 | 18,493,552 | <0.001 | 0.998 |
| | Low Limit | 318,435,844 | 1000 | 318,435,844 | 18,493,552 | <0.001 | 0.998 |
| Time × Group | Sphericity | 6,484,594 | 1 | 6,484,594 | 376,601 | <0.001 | 0.891 |
| | Greenhouse-Geisser | 6,484,594 | 1000 | 6,484,594 | 376,601 | <0.001 | 0.891 |
| | Huynh-Feldt | 6,484,594 | 1000 | 6,484,594 | 376,601 | <0.001 | 0.891 |
| | Low Limit | 6,484,594 | 1000 | 6,484,594 | 376,601 | <0.001 | 0.891 |
| Error(time) | Sphericity | 792,063 | 46 | 17,219 | | | |
| | Greenhouse-Geisser | 792,063 | 46,000 | 17,219 | | | |
| | Huynh-Feldt | 792,063 | 46,000 | 17,219 | | | |
| | Low Limit | 792,063 | 46,000 | 17,219 | | | |

**Table 4.** Effect size between subjects.

| | Sum of Type III Squares | df | Quadratic Mean | F | *p* | Partial Eta Square |
|---|---|---|---|---|---|---|
| Intercept | 922,180,010 | 1 | 922,180,010 | 32,145,201 | 0.000 | 0.999 |
| Group | 6,288,844 | 1 | 6,288,844 | 219,215 | 0.000 | 0.827 |
| Error | 1,319,646 | 46 | 28,688 | | | |

## 4. Discussion

The pedagogical potential of digital games for language acquisition has gained recognition among educators [28]. Digital game-based language learning refers to the use of

computers with an identifiable educational presence to enhance specific aspects of language proficiency [29]. Recognizing the potential of digital games to enhance intrinsic motivation and enjoyment in language learning, many teachers have incorporated them into their language learning curricula [30]. This study aimed to examine the impact of the popular digital game-based learning platform "Kahoot!" and various game-based activities on the acquisition of basic Italian grammar among Ukrainian children, as measured by vocabulary expansion in pre- and post-tests over a 4-week period. The results revealed a significant increase in Italian vocabulary in both conditions after the 4-week intervention, with a larger effect observed in the game-based condition. These findings align with previous studies demonstrating the positive effects of mobile devices (digital game-based learning) on language acquisition, particularly in writing and listening skills [31,32]. Kahoot! is merely one of the most prevalent platforms employed for pedagogical purposes. Specifically, there exist alternative platforms that can be utilized for foreign language acquisition as well.

Mobile applications [33], Socrative [34], or Duolingo [35] represent a few of the tools employed in this regard.

Klimova [33] indicated that the use of mobile phones and/or smartphones and their apps generate positive effects as far as English learning is concerned. The findings show that mobile applications have a primary impact on the development of all language skills, including their testing. Nevertheless, these applications especially have quite a significant impact on the development and retention of students' vocabulary.

Another effective tool is "Socrative", which is a smart student response system that enables instructors to discover or assess what students have learned in their lectures in real time. The author indicated that Socrative is a right tool that can help to improve users' engagement in the classroom [34].

Lastly, the famous language learning application "Duolinguo", and its effectiveness, is discussed [35]. Loewen and colleagues [35] stated that the effectiveness of the application depends on how the user uses it (alone or with a teacher). In addition to this, the phonological and morphosyntactic structure of the language being learnt is also a variable to be taken into account.

Indeed, various challenges arise in the context of Ukrainian individuals acquiring proficiency in the Italian language. As gleaned from the analysis by Retaro [36], those hailing from Ukraine encounter difficulties not solely at the morphosyntactic level, but also phonologically. This phenomenon stems from the substantial disparity between the sound systems inherent in the Italian and Ukrainian languages. Nevertheless, it appears that these challenges tend to diminish during early stages, as the solidification of language pronunciation takes place within an early timeframe.

Despite the above-mentioned difficulties, our results show a positive effect with respect to learning the Italian language.

Research on language learning through games has predominantly focused on vocabulary expansion [37]. Indeed, given the significance of syntax and lexicon in teaching a foreign language, a wide array of games has been meticulously devised to foster students' enthusiasm and active participation during their lessons. Evidently, as asserted by Cárdenas-Moncada et al. [38], incorporating games into grammar teaching, and especially into vocabulary teaching, proves highly beneficial, not merely in enhancing students' comprehension but also in fostering an animated and dynamic learning environment within the classroom [19].

This was evident in both conditions, but more pronounced in the game-based condition. This can be attributed to the activities, particularly those utilizing the digital platform, enhancing the children's ability to retrieve vocabulary from memory. The speed at which this retrieval occurs remains unclear, as rapid online vocabulary retrieval and maintenance in working memory are crucial [39]. Moreover, the increased motivation and reinforcement associated with playful activities compared to traditional classroom teaching could explain these findings. Previous studies have reported improved motivation, engagement, and cognitive abilities among students benefiting from a playful teaching approach [11].

Additionally, games, particularly digital games, have been linked to greater engagement of cognitive functions such as working memory, inhibitory control, and cognitive flexibility. While the exact extent of their impact remains uncertain, it is worth exploring the development of gamified cognitive tasks in the future [40]. Gaming plays a crucial role in developing skills and soft abilities due to its experiential approach, intrinsic motivation, and structural organization. The collective aspect of gaming also holds significance, as the internal dynamics of a group influence its composition and are intrinsically linked to the employed leadership style [41]. Furthermore, motivation is a key factor in successfully engaging participants in data gathering or behavioural change. Evidence suggests that lack of participant motivation negatively affects data quality, with tasks often perceived as monotonous and repetitive, resulting in reduced attention [42].

## 5. Conclusions

Our research aimed to examine the impact of game-based learning on the development of foreign language vocabulary. It is important to note that this study has certain limitations, including a small sample size, limited variables, and the need for further investigation. In order to provide a comprehensive analysis of foreign language acquisition, future studies should consider incorporating additional indicators such as complexity, accuracy, and fluency. Furthermore, the study did not explore the potential effects of the game-based learning approach on learner motivation, attitudes, and enjoyment. Lastly, a follow-up study is necessary to determine the strength of the observed learning outcomes. Despite these limitations, our findings contribute to the existing body of literature and offer valuable insights into this understudied area. Moreover, it is important to consider the specificity of the sample and the historical context that Ukrainian people are experiencing. Beyond learning Italian, a play-based educational approach would decrease perceived stress and make the educational environment in which these children find themselves more inclusive and positive.

**Author Contributions:** Conceptualization, A.F.; methodology, C.E.; software, F.C.; validation, F.C. and C.E.; formal analysis, M.R.; investigation, F.B.; resources, F.C. and C.E.; data curation, M.R.; writing—original draft preparation, A.F., C.E. and M.R.; writing—review and editing, A.F., M.R. and F.B.; visualization, M.R.; supervision, A.F. and F.B.; project administration, A.F. All authors have read and agreed to the published version of the manuscript.

**Funding:** This research received no external funding.

**Institutional Review Board Statement:** The study was conducted in accordance with the Declaration of Helsinki. The study was approved by the Ethics Committee and the Academic Senate of the Rome University of International Studies.

**Informed Consent Statement:** Informed consent was obtained from all subjects involved in the study.

**Data Availability Statement:** The data presented in this study are available on request from the corresponding author.

**Conflicts of Interest:** The authors declare no conflict of interest.

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
