# Peer review of "Learning Italian as a Second Language in a Sample of Ukrainian Children: A Game-Based Learning Approach"

_pediatrrep, doi:10.3390/pediatric15030046_

Round 1
Reviewer 1 Report
The work investigates the effect of game-learning approach on foreign language vocabulary acquisition in a small sample of Ukraine children. The topic of the effects of game-learning is of interest, nonetheless the work presents some unclear points/ limitations that should be taken into account.
Title: I suggest declare that this is a pilot study
Abstract
Line 19. I suggest changing the sentence “Children were involved in Italian learning 3 hours…”. It is not clear, was it three hours twice a week? For a total of 6 hours a week? Please reformulate
Line 23. Do you mean scares literature on foreign language learning through video-games? Please be clearer for the reader
Introduction
In this section, the authors summarize recent results on the effects of game-learning approach. A login list of effects of several variables is reported. Nonetheless, no space is given to the interpretation that the authors of the studies reported have given to their results, for example on the mechanism which are at the basis of these effects. I suggest giving space to this point.
Moreover, only one study is referred which investigated this topic in second language learning. If this is the only existing study, it is necessary to declare it. If this is not the case, please deepen this literature review.
Finally, the aim of the study is reported, but not any hypothesis on the results. Please add some hypothesis (or declare why you don’t have one).
Materials and method
The participants are Ukrainian refugees. Why didn’t you measure the possible presence of PTSD? If present, PTSD could have a strong effect on learning.
Line 93. Eliminate “to assess these children”.
Line 94. Economic Social Scale (SES), please insert the range of the possible scores, to be clearer for the reader
Line 103. Again, it is not clear to me: three hours per day, but 6 per week?
MacArthur-Bates CDI (Communicative Development Inventory). Why did you include the CDI, which is an instrument developed for children from 8 to 30 months of age?
You describe both the versions of the CDI, but you don’t say which did you use and why. Moreover, who complete the questionnaire? The children’s parent? This could be a problem, considering that they are not Italian mother-tongue. Moreover, did they complete the Italian version of then CDI, or an Ukraine version of it? It is not clear.
K-SADS. Who complete the questionnaire for each child?
Line 150. Again, it is not clear to me: three hours per day, but 6 per week?
Line 175-177 “This incentive system fueled the students' motivation to win and encouraged them to pay closer attention during the explanations, leading to better comprehension and improved performance in subsequent kahoots”. Do you have data about this or it’s your qualitative impressions? If you have data, report them in the paper. If you don’t have data, declare that this is your impression.
Results
Line 205. Report the p value, not just < .05, and the effect size
Please report all the effects, not only the interaction effect. Report all the values, in table 1 or in the text (F values, p values and effect size).
Discussion
Line 219-220 “This study aimed to examine the impact of the popular digital game-based learning platform "Kahoot!"”, it was not clear for me, till the discussion, that you were testing Kahoot!. It seemed to me that you were testing the effect on vocabulary learning of game-based learning, not of Kahoot in particular. Please make this point clear for the reader from the aims.
Line 225. “effects of mobile devices”. Not clear to me: do you want to test the effects of game-based learning or of mobile devices? Because it is not the same.
General: which could be the cognitive mechanisms at the basis of the effect of game-learning approach on vocabulary learning? It’s not clear for me the authors idea. Moreover, as a future step of the research, how this point could be investigated?
Please revise the English form of the final version of the paper
Author Response
Dear reviewer,
My colleagues and I deeply appreciate the corrections and the time you have dedicated to reading and revising the manuscript. We are delighted that you found it interesting and well-done. Despite this, we have added sections that address your comments, and in the attached file, we have attempted to provide responses to the issues you raised. We hope that the revised manuscript will be satisfactory. Thank you for the support and assistance you have provided.
Kind Regards

Reviewer 2 Report
This well-written paper describes a sound pedagogical approach to teaching foreign languages. The basic premise, i.e., that game-based learning can be pedagogically effective, is fairly uncontroversial, and the literature review supports it well. In view of this, a question can be asked what original information the paper presents. Futhermore, the low numbers of participants and multitude of confounding variables that have not been controlled for cast doubts on the validity of the study - this is less of a problem because the authors are arguing for a point that is uncontroversial. On the whole, I do not see anything that might hinder the publication of this paper, but I fail to see how it adds to existing scholarship.
Author Response

(The authors gave the same response as above.)

Reviewer 3 Report
The perspective of the research is adequate and meets a current need, to provide appropriate training for Ukrainian children and to improve their reception in different countries of Europe.
The development of the research is well structured. Although the sample is minimal, the structure of the didactic proposal and the research design is correct. Several resources are presented in the methodology to analyze the learning, but the presentation of the results is quite brief. It is recommended to expand this section with some other data confirming the learning of the experimental group following a game-based learning methodology. It is suggested to include some more tables with other data.
In addition, if the tools and activities in Kahoot are available openly, it is suggested to offer some online example, to contrast the experience and even to be able to replicate it with other learners or with other host languages.
It is also advisable to review other mobile tools and applications (Klimova, 2018) to compare possibilities, such as Socrative (Kaya, & Balta, 2016; Saracoglu, & Kocabatmaz 2019; Vasconcelos, & Balula 2017) or Duolinguo (Loewen, et al. 2019).
In addition, there is no reference to the specific issues of learning Italian for Ukrainians and the current challenges in this regard. It would be advisable to include some current reference (Retaro, 2023; Scolaro & Tomasi, 2023).
Finally, the bibliography is correct, but less than 30% is from the last 5 years. Several suggestions are provided to expand the theoretical framework of this research.
Suggested investigations
Retaro, Valentina (2023) La fonetica nell’apprendimento dell’italiano l2 in apprendenti adulte di origine ucraina: riflessioni teoriche e proposte didattiche., Italiano Linguadue, v.15 n.1 https://doi.org/10.54103/2037-3597/20408
Scolaro, Silvia & Tomasi, Matilde (2023). Apprendenti ucraini di italiano l2 in progetti di accoglienza: case-study. Italiano Linguadue, v.15 n.1 https://doi.org/10.54103/2037-3597/20407
Kaya, A., & Balta, N. (2016). Taking advantages of technologies: using the socrative in english language teaching classes. International Journal of Social Sciences & Educational Studies, 2(3), 4-12.
Klimova, B. (2018). Mobile phones and/or smartphones and their apps for teaching English as a foreign language. Education and Information Technologies, 23(3), 1091-1099. https://doi.org/10.1007/s10639-017-9655-5
Loewen, S., Crowther, D., Isbell, D., Kim, K., Maloney, J., Miller, Z., & Rawal, H. (2019). Mobile-assisted language learning: A Duolingo case study. ReCALL, 31(3), 293-311. https://doi.org/10.1017/S0958344019000065
Saracoglu, G., & Kocabatmaz, H. (2019). A study on Kahoot and Socrative in line with preservice teachers’ views. Educational Policy Analysis and Strategic Research, 14(4), 31-46. https://doi.org/10.29329/epasr.2019.220.2
Vasconcelos, S. V., & Balula, A. (2017). Socrative: Using mobile devices to promote language learning. En M. Mills, & D. Wake (Eds.), Empowering Learners With Mobile Open-Access Learning Initiatives (pp. 215-237). IGI Global. https://doi.org/10.4018/978-1-5225-2122-8.ch012
Author Response

(The authors gave the same response as above.)

Reviewer 4 Report
Thank you for allowing me to read and review your article.
Strengths - you did a very thorough job selecting and qualifying the participants. Also, the procedure ad the statistical tool was selected appropriately for this study.
What can be improved - there are a few statements that should be revised or deleted, such as saying there isn't a lot of research in FLL...or at the end of the discussion where you started talking about motivation (something that was not part of your research design).
Please see my comments on the attachment. The comments may need to be opened in the pdf to view them.

Overall, the language is fine. It could be a bit more sophisticated.
Author Response

(The authors gave the same response as above.)

Round 2
Reviewer 4 Report
I am happy with most of the changes. There are two small things I would like to see modified.
1) in Table 2, it shows the p values as 0.000. This should say the exact number or < .001
2) The tables are not formatted to APA7, so you just need to select the table format with no column lines.
Otherwise, this looks good to me.
Author Response
Dear Reviewer,
thank you for your further suggestions. The corrections have been made.
We hope that this latest version is correct in form and content.
Thank you very much for your support.
Kind Regards
